# Hepatitis C Virus among Female Sex Workers: A Cross-Sectional Study Conducted along Rivers and Highways in the Amazon Region

**DOI:** 10.3390/pathogens8040236

**Published:** 2019-11-14

**Authors:** Aldemir B. Oliveira-Filho, Diego Wendel F. Aires, Natalia S. Cavalcante, Nairis Costa Raiol, Brenda Luena A. Lisboa, Paula Cristina R. Frade, Luana M. da Costa, Luiz Marcelo L. Pinheiro, Luiz Fernando A. Machado, Luisa C. Martins, Gláucia C. Silva-Oliveira, João Renato R. Pinho, Emil Kupek, José Alexandre R. Lemos

**Affiliations:** 1Instituto de Estudos Costeiros, Universidade Federal do Pará, Bragança PA 68600-000, Brazil; diego_wendel_@hotmail.com (D.W.F.A.); nattycavalcante2014@hotmail.com (N.S.C.); nayraiol94@gmail.com (N.C.R.); brenda_ufpa1@yahoo.com (B.L.A.L.); gcoliveira@ufpa.br (G.C.S.-O.); 2Programa de Pós-Graduação em Saúde Coletiva, Universidade Federal de Santa Catarina, Florianópolis SC 88040-900, Brazil; emil.kupek@ufsc.br; 3Núcleo de Medicina Tropical, Universidade Federal do Pará, Belém PA 66055-240, Brazil; paulacrfrade@gmail.com (P.C.R.F.); luanam.c@hotmail.com (L.M.d.C.); caricio@ufpa.br (L.C.M.); 4Faculdade de Ciências Biológicas, Campus do Marajó, Universidade Federal do Pará, Soure PA 68870-000, Brazil; lmarcelo@ufpa.br; 5Instituto de Ciências Biológicas, Universidade Federal do Pará, Belém PA 66077-830, Brazil; lfam@ufpa.br (L.F.A.M.); jalemos@ufpa.br (J.A.R.L.); 6Instituto de Medicina Tropical, Universidade de São Paulo, São Paulo SP 05403-000, Brazil; jrrpinho@usp.br; 7Departamento de Saúde Pública, Universidade Federal de Santa Catarina, Florianópolis SC 88040-900, Brazil; 8Centro de Hematologia e Hemoterapia do Pará, Laboratório de Biologia Molecular, Belém PA 66033-000, Brazil

**Keywords:** epidemiology, HCV, genotype, risk factors, protease inhibitors, female sex workers, women’s health, social vulnerability, amazon region

## Abstract

Background: Previous studies found a high prevalence of pathogens among female sex workers (FSWs) in the Amazon region, and established their parenteral and sexual transmission. This study estimated the prevalence of hepatitis C virus (HCV) infection and associated risk factors, and the frequency of HCV genotypes and resistance-associated substitutions (RASs) in this vulnerable group. Methods: Distinct sampling methods were used to access 412 FSWs in cities and riverside communities in the Amazon region from 2015 to 2018. Three methods for HCV diagnosis were used to determine infection status. HCV genotypes and RASs were identified by sequencing and nucleotide fragment analysis. An association between HCV infection and exposure factors was determined by bivariate and multivariate analysis. Results: In total, 44 (10.7%) FSWs were exposed to HCV, and 32 (7.8%) of them had active infection. Nine socioeconomic characteristics and risky sexual behaviors were associated with HCV exposure, particularly unprotected sex and condom exemption for the clients who paid extra money. Genotype 1 (81.3%) and 3 (18.7%) were detected. The frequency of FSWs with RASs was 23.1% (6/26) for grazoprevir related to the occurrence of substitutions Y56F and S122G. Conclusions: HCV infection among FSWs is highly prevalent and dominated by genotype I. Urgent preventive and treatment measures are required to reduce HCV infection in FSWs and the general population.

## 1. Introduction

The hepatitis C virus (HCV) infection is a major public health problem worldwide; an estimated 100 million people have serological evidence of HCV exposure, which may be associated with a growing trend in deaths per year [1]. Many cases of acute infection have no symptoms and remain undiagnosed, with later progression to chronic liver disease, cirrhosis, liver failure, or hepatocellular carcinoma [2,3]. Blood transfusion, contaminated blood products, and unsafe medical practices were the main routes of global HCV spread until the early 1980s [4]. Currently, illicit drug use is a major risk factor for HCV infection [2,3]. Risky sexual behavior may also be a risk factor for HCV infection, and has been reported as an important route of viral transmission among people who use illicit drugs (PWUDs), female sex workers (FSWs), and men who have sex with other men (MSMs) [5,6,7].

A high genetic diversity of HCV has been detected. Currently, these viruses can be classified into seven genotypes (1–7) and numerous subtypes (1a, 1b, 1c, 2a, 2b, 2c, 3a, 3b, 4a, 5a, and others). HCV genotypes and subtypes may vary in geographic distribution, risk factors associated with viral exposure, and treatment response [8,9]. Genotypes 1, 2, and 3 have already been detected in several countries. A high prevalence of genotype 1 has been reported in different population groups, and this genetic variant of HCV has shown the worst response to antiviral therapy [1,8,9]. Thus, efforts were made to improve the efficacy and tolerability of HCV treatment, leading to the development of multiple direct-acting antivirals (DAAs) [9,10]. These drugs are molecules that target specific nonstructural proteins of the virus and results in disruption of viral replication and infection, regardless of HCV genotype [9,10,11]. Studies have indicated that protease inhibitors induce rapid selection of resistant viral variants, but the combination of protease inhibitors with other drugs (e.g., interferon-alpha and ribavirin) has indicated a significant improvement in the sustained virological response rate [11,12,13]. In Brazil, drugs, such as boceprevir, telaprevir, simeprevir, and sofosbuvir, are provided at no cost to people infected with HCV who have liver fibrosis or cirrhosis [14].

In this South American country, it is estimated that 1% to 2% of the general population is infected with HCV, and genotypes 1, 2, and 3 have been recorded, but there are variations according to the geographical region [15,16,17,18]. Furthermore, resistance-associated substitutions (RASs) located in NS3, such as V36L, V55A, Q80L, and R155K for subtype 1a, and T54S, V55A, R117H, and D168G for subtype 1b, have already been identified in HCV-infected Brazilians [19]. In the Amazon region (northern Brazil), several epidemiological studies indicate a high prevalence of HCV infections and the predominance of genotype 1 in different population groups, such as indigenous people, people living in riverside communities, blood donors, patients undergoing hemodialysis, patients with multiple blood transfusions, and PWUDs [5,18,20,21,22,23,24,25]. Factors associated with parenteral and sexual exposure to HCV have been detected, including: Shared use of manicure and pedicure instruments, use of home-sterilized needles and syringes, unprotected sexual intercourse, more than 12 sexual partners, daily drug use, drug use for more than three years, and shared use of drug paraphernalia [5,26,27,28,29]. However, much of the epidemiological information on HCV infection among other vulnerable groups in the Amazon region, such as FSWs, is still unknown.

In general, FSWs may act as reservoirs of sexually transmitted infections (STIs) and have an increased potential for the dissemination of pathogens to their partners and clients, contributing to the infection spread in the general population [30,31]. Low adherence to condom use, multiple sexual partners, unsafe sexual practices, illicit drug use, presence of genital ulcerative disease, and co-infection with other STIs increase the risk of HCV transmission [31,32,33,34]. In the state of Pará (Amazon region), a high prevalence of hepatitis B virus infections (13.7%) and *Treponema pallidum* (36.9%) were detected in FSWs; the consumption of illicit drugs, unprotected intercourse, exemption of condoms for regular clients or those paying an extra fee, and more than five years of sex trade were all shown to be risk factors for acquiring these pathogens [30,35]. Although best known for its extraordinary biological diversity and environmental importance to the world, the Amazon region is also an underdeveloped area, with high levels of poverty, limited transport infrastructure, and inadequate health services, thus allowing the spread of various pathogens and the occurrence of numerous infectious diseases [30,35,36,37,38]. Based on that, this study assessed FSWs in different locations in the Amazon region to estimate the prevalence of infections and factors associated with HCV exposure, as well as to determine the frequency of HCV genotypes and to assess the presence of RASs, aiming to provide consistent information for targeting control measures and prevention.

## 2. Results

### 2.1. Study Sample

In total, this study assessed 482 FSWs in different points of the sex trade in the Brazilian state of Pará in the Amazon region. However, 70 FSWs were excluded from the sample due to being under 18 years of age, signs of the effect of alcoholic beverages during a meeting with researchers, and solicited financial resources to provide biological samples and personal information. Thus, this study sample consisted of biological samples and personal information provided by 412 FSWs (Figure 1), of which 180 women worked in the sex trade points (bars, squares, ports and stations of river fuels) in 7 municipalities and 18 riverine communities of the Marajó Archipelago, and the other 232 women worked in the sex trade points (bars, restaurants, parks, gas stations, social event venues, and truck parking spots) near highways, in 11 municipalities in the state of Pará (Appendix A). In the municipalities of Salinópolis and Terra Alta, FSWs were accessed for convenience (COS) due to difficulties encountered in developing other methods, such as respondent-driven sampling (RDS), Time location sampling (TLS) and take-all sampling (TAS) were performed elsewhere in this study. Furthermore, there were no comparisons between FSWs working in municipalities and riverside communities because the latter reported that they had also offered sexual services in the municipalities of the Marajó Archipelago and others, especially during summer holidays and festivities in the cities of Belém, Breves, Marapanim (Marudá District), Salinópolis, and Soure.

### 2.2. Characteristics of FSWs

Most women reported being born in the state of Pará (77.2%). The other FSWs reported being born in other Brazilian states, such as Amapá (18/412–4.4%), Amazonas (3.9%), Maranhão (6.8%), Tocantins (4.4%), and Piauí (3.3%). The mean age was approximately 26.5 ± 6.5 years (range 18 to 47 years). Most of them reported being single, heterosexual, non-white, low levels of education, and low monthly income (Table 1). The monthly income of the FSWs ranged from 400 to 2100 Brazilian reals, although most (74.5%) reported a monthly income of approximately one minimum wage (R$ 890 in Brazilian currency, equivalent to US$ 220 with the exchange rate of approximately 4 R$ for one US$). The FSWs reported that a sexual encounter cost between R$ 20 and 100, with an average price of about R$ 30 (equivalent to US$ 7.42). A typical basic sexual program included kisses, caresses, and vaginal sex. Sexual fantasies, oral sex, and anal sex could be included in the program for an extra payment. Most women (45.6%) reported having been working as a sex worker for more than seven years and having suffered frequent episodes of physical (58.5%) and sexual (37.4%) violence in the sex trade. Some of them also reported having sex without condoms, especially with clients who paid more for the sexual program (31.8%) and with regular clients (18.4%).

### 2.3. Prevalence of Infections and Frequency of HCV Genotypes

In total, 44 (10.7%) FSWs had anti-HCV antibodies by enzyme immunoassay (EIA). Among these, 32 (7.8%) FSWs also featured HCV RNA, indicating active HCV infection. Conversely, 12 (2.9%) FSWs had anti-HCV antibodies by EIA and immune blot, with the absence of HCV RNA by real-time PCR, indicating non-active HCV infection, and presented with presumed spontaneous clearance of HCV (Table 2). None of the 44 (10.7%) FSWs exposed to HCV were aware of their HCV infection status (active or inactive) and received no treatment. Furthermore, the HCV genotypes were determined by NS5b amplification, with successful sequencing in all 32 HCV RNA-detectable samples. Genotypes 1 (81.3%) and 3 (18.7%) were identified (Table 2). These samples belong to subtypes 1a (34.4%), 1b (46.9%), 3a (15.6%), and 3b (3.1%) (Appendix A). The results of the genotyping by phylogenetic analysis or the online analysis tool assigned the same genotype.

### 2.4. Factors Associated with HCV Exposure

The bivariate analysis identified the following factors associated with HCV exposure: (i) Up to elementary school, (ii) up to one minimum wage, (iii) illicit drug use, (iv) unprotected sex, (v) more than five sexual partners, (vi) condom exemption for clients paying extra, (vii) more than seven years working in the sex trade, (viii) changes in genitalia (wart, wound, and/or itching), and (ix) did not perform medical/gynecological examination (Table 3). The multivariate analysis identified an association of the same nine factors with HCV exposure. The Hosmer–Lemeshow test indicated that the final model (_HL_χ^2^ = 10.6; *p* = 0.4) had a good fit. The factors most strongly associated with HCV exposure (aOR > 10) were unprotected sex and condom exemption for the clients who paid extra (Table 3). Factors not associated with HCV exposure are shown in the Appendix A. Also, no association was detected between genotypes 1 and 3 and the FSWs’ epidemiological characteristics (Appendix A).

### 2.5. Substitutions Associated with Resistance

Only genotype 1 samples (n = 26) were analyzed by amplification and sequencing of the NS3 viral protease coding region, which was successful in all cases. The frequency of FSWs with RASs was 23.1% (6/26) for grazoprevir. Two RASs were detected: S122G (5/26, 19.2%) and Y56F (1/26, 3.9%). All RASs were detected in FSWs with subtype 1b. Interestingly, modification N174S was detected in six FSWs with subtype 1a. According to geno2pheno, N174S may indicate reduced susceptibility to telaprevir. In total, three modifications were found in the sequences of the NS3 viral protease coding region (Appendix A).

## 3. Discussion

This study is the first epidemiological investigation on HCV infection among FSWs in the Brazilian Amazon region. It revealed important characteristics of FSWs regarding HCV infection, which should be used to promote the health of these women and to direct control and prevention measures against this infection.

Overall, the socio-demographic and economic profile of FSWs in this study is consistent with the characteristics of FSWs accessed in other Brazilian cities and in different locations around the world [6,39,40,41,42,43,44,45]. Poor education and low monthly income are clear indications of the vulnerability of women who offer sexual services on highways and rivers in the Amazon region. These characteristics indicated that the need for better financial resources, the lack of knowledge how to seek better health conditions (such as self-care, self-worth, and medical care when necessary), and neglect in health risk situations (such as illicit drug use and inconsistent condom use) could facilitate the acquisition and spread of pathogens, such as HCV. This epidemiological scenario is aggravated by the migration of FSWs in search of customers and the provision of sexual services in different locations in this Brazilian region, thus enhancing the spread of pathogens in the general population.

In the Amazon region, the estimated prevalence of HCV infection ranges from 1% to 3%, with higher rates being recorded in vulnerable groups, such as PWUDs (23.1%), patients undergoing hemodialysis (8.4%), and patients with hemophilia (48.4%) [5,20,21,22,23,24,29,36,46]. The prevalence of HCV infection detected in this study was 10.7%, and most of these infections were classified as active (7.8%). This value is much higher than the prevalence of HCV infection (0.9%) recently found in 4154 FSWs in 12 metropolitan areas of Brazil [6]. Such a high HCV prevalence among FSWs in the Amazon region can be the result of viral spread by sexual intercourse, a hypothesis strongly supported by factors associated with HCV exposure detected in the present study.

Sexual transmission of HCV is well known in high-risk groups, such as MSM, PWID, and human immunodeficiency virus (HIV)-infected individuals [47,48,49,50,51]. The sexual transmission of HCV is supported by the isolation of HCV RNA from semen and cervical smears [33,52,53,54], and also by the investigation and genetic identification of the same HCV strain in some cases of infected persons who had unprotected sex [32,33,55,56,57]. Furthermore, other studies have shown that HCV can be transmitted more efficiently through the association between unprotected sex, mucosal trauma, and the presence of genital ulcerative disease [32,33,34]. Most of these factors were detected in the present study, with an additional burden provided by unfavorable socioeconomic conditions, thus underlying the need for effective control and prevention measures.

Among the various interventions needed, the first point is the provision of efficient health services for the entire population of the Amazon region. Improvement and availability of resources, diagnostics, and technical skills for infectious diseases are essential, including the availability of testing and treatment for HCV-infected individuals and the active search for infection in vulnerable groups. The rapid point-of-care test methods for identifying important pathogens, such as human immunodeficiency virus (HIV), HCV, HBV, and *T. pallidum*, should be scaled up, especially for vulnerable groups. This would make it possible to identify a greater number of people infected with HCV, many of whom are unaware of their carrier status.

Another urgent need is the provision of efficient treatment for HCV-infected people based on genotyping and RASs assessment. Preferably, DAAs with a proved effectiveness of >90% over a 12-week treatment course should be provided. For this to happen, the public health services in this Brazilian region should offer more complex genotyping and assessment of RASs. Preferably, DAAs should be made available, as these have shown over a 90% effectiveness in a 12-week treatment schedule [11,48]. In this study, there was a predominance of genotype 1 (81.3%), especially subtypes 1a (34.4%) and 1b (46.9%), with the presence of RASs in antiviral therapy-naive FSWs. Historically, epidemiological studies conducted in the Amazon region have indicated the predominance of HCV genotype 1 but without an evaluation of the presence of RASs [5,18,20,36]. Studies conducted in other Brazilian regions reported the presence of genetic variants in chronic HCV-infected patients and not treated with DAAs, which may vary from 3.2% to 18.9% [19,58,59,60]. In the Brazilian state of São Paulo, 16 out of 125 HCV-infected blood donors (12.8%) had RASs for boceprevir, telaprevir, and simeprevir [19]. In this study, this rate was even higher as 6 of 26 (23.1%) HCV-infected FSWs had RASs for grazoprevir. The S122G and Y56F substitutions were identified, and have already been described in the scientific literature [61,62,63]. On the other hand, there is no scientific report of the association of N174S with resistance in non-responders to viral therapy, but there are reports of its presence in patients infected with HCV subtype 1a [64]. Overall, the HCV genetic information reported in this study should serve as a warning to health authorities and reinforce the urgent need for measures to reduce HCV transmission.

Also, the results indicate the need for targeted prevention concerning risks related to HCV and other pathogens, such as HIV, HBV, and *T. pallidum*. The main factors associated with HCV exposure among FSWs could be related to sexual behavior and drug use. Targeted education of this high-risk group and improved condom availability/distribution would be a natural starting point of preventive actions. Considering the characteristics of FSWs, the execution of educational activities directed to self-assessment, self-esteem, self-confidence, knowledge acquisition, social benefits, and social inclusion can modify STIs’ risk perception and hence promote safer health choices for these women in the Amazon region. According to Frade et al. [30], community health workers can initiate these actions by seeking, guiding, and directing the FSWs to participate in conversation circles and workshops in small health centers in municipalities and riverside communities. Other strategies and actions may facilitate access to public health services, bring care to people with reduced education who need guidance, and stimulate self-care, self-worth, the appropriation of rights, and self-knowledge [30,60]. Finally, health authorities in the Amazon region and other underdeveloped areas should seek. appropriate strategies and actions to effectively identify and treat current infections, as well as prevent the emergence of new infections.

The present study has several limitations. The first factor is the age limit criterion (18 years) applied to eligible participants, since FSWs reported beginning to exchange sex for money during early adolescent years (<18 years). Interview data are self-reported, and some information, such as drug use or sex-related risk behaviors, may contain response or recall bias; this fact is important and should be considered. In addition, recent infections may not have been detected by HCV screening for EIA, as small antigen/antibody concentrations (below the detection limit of the test) may have been diagnosed as negative. Lastly, the causality may be limited by the cross-sectional study design.

## 4. Materials and Methods

### 4.1. Study Design and Participants

This cross-sectional study collected biological samples and personal data from FSWs working in 18 towns and 18 riverine communities in the Brazilian state of Pará between January 2015 and July 2018 (Figure 1). The towns of Breves and Santa Maria do Pará were considered key locations for the present study due to several factors. Breves is the largest town of the Marajó Archipelago (one of the world’s major estuarine archipelagos), located at the intersection of two important river routes (Belém-Macapá and Belém-Santarém). It is a regional center of the local lumber industry and therefore an attractive place for intense sex trade [30]. Santa Maria do Pará is the meeting point of the two main highways in the state of Pará, with an intense circulation of products and people, which makes it a strategic place for sex trade [35]. Given this, most collection points were located in or near the towns of Breves and Santa Maria do Pará and were reached by car, ship, or boat. Four sampling methods were used: Respondent-driven sampling (RDS), time location sampling (TLS), take-all sampling (TAS), and convenience sampling (non-probabilistic) (COS) [65]. Each sampling method was employed to access the largest possible number of FSWs in the study area.

Blood samples were collected by venipuncture using disposable syringes and tubes containing anticoagulant [30]. Personal data were collected in direct interviews using a standardized questionnaire that focused on the social, demographic, and economic characteristics of the participants, and their working conditions and health [30,35]. All blood samples obtained were stored and transported to laboratories in the cities of Bragança and Belém to perform laboratory procedures.

The criteria for the inclusion of sex workers in the sample were: (i) Being of the female sex (biologically), (ii) having had sex in exchange for money in the Amazon region during a period of at least three months, (iii) being at least 18 years of age, and (iv) being able to provide a biological sample and personal epidemiological details (as well as providing informed consent). FSWs were excluded from the sample when they (i) were under the influence of psychotropic drugs during any meeting with researchers or (ii) if the researcher judged that they may pose a risk to his/her physical integrity.

### 4.2. Laboratory Tests

All samples collected were tested for the presence of anti-HCV antibodies using enzyme immunoassay (EIA—Murex anti-HCV 4.0, DiaSorin). Nonreactive samples were classified as HCV negative, and no additional tests were performed on these samples. Reactive samples for anti-HCV antibodies were further evaluated by quantitative real time reverse-transcription polymerase chain reaction (RT-qPCR) and Immuno Blot Assay to confirm HCV infection status [5,36]. In this study, the presence of anti-HCV antibodies by EIA (confirmed by the positive result using PCR or Immunoblot) was indicated as a laboratory marker of HCV exposure [5,36].

### 4.3. Statistical Analysis

All study data generated were entered into an Excel database and converted to SPSS. Chi-square and Fisher’s exact test were used to evaluate associations between risk factors and the outcome (HCV-infected: yes/no). Odds ratios (OR) and 95% confidence intervals (CI) were used as measures of the strength of association between the outcome and independent variables. Independent variables associated with the outcome with *p*-values < 0.2 in bivariate analysis were entered into a backward stepwise multivariate logistic regression. The latter used a 0.05 significance level for the type I error. The overall fit of the final model was assessed using the Hosmer–Lemeshow test. All statistical analyses and procedures were performed using SPSS 20.0 for Windows.

### 4.4. HCV Phylogenetic Analysis and Genotyping

Only positive samples by real-time PCR (cDNA HCV) were subjected to nested PCR to amplify a 400-bp fragment from the non-structural region 5B (*NS5B*) gene, including part of the RNA-dependent RNA polymerase, using previously described PCR primers and conditions [16,66]. The PCR product was purified using the QIAquick kit (QIAGEN, Hilden, Germany) and directly sequenced using the Big Dye Terminator Cycle Sequencing Ready Reaction (Applied Biosystems, Foster City, CA, USA) according to the manufacturer’s procedure. The sequences were edited and aligned using AliView software [67].

The genotypes were determined using phylogenetic analyses with HCV reference sequences of HCV genotypes, obtained from the National Center for Biotechnology Information (https://www.ncbi.nlm.nih.gov) and later confirmed by COMET (https://comet.lih.lu/index.php?cat=hcv). To verify the clustering of HCV sequences, a maximum-likelihood (ML) phylogenetic tree was reconstructed with the PhyML 3.1 [68] under the best nucleotide substitution model, selected by the SMS (Smart Model Selection) software [69] integrated into the PhyML Web server. The heuristic trees search was performed using the SPR branch-swapping algorithm, and the branch support was calculated with the approximate likelihood-ratio (aLRT) SH-like test [70]. The tree was drawn with FigTree 1.4.4 (http://tree.bio.ed.ac.uk/software/figtree).

### 4.5. Analysis of Mutations Associated with Resistance to Protease Inhibitors

All the genotype 1 samples were subjected to nested or semi-nested PCR to amplify the region encoding the non-structural region 3 (NS3) protease catalytic domain and sequenced as described above [71,72,73,74]. Identification of mutations and predictions of phenotypic resistance of the NS3 region to different drugs were determined using the online tools geno2pheno [hcv] (http://hcv.Bioinf.mpi-inf.mpg.de/index.php) and HCV-GLUE (http://hcv-glue.cvr.gla.ac.uk).

### 4.6. Availability of Data

The HCV sequences from this study were deposited in GenBank under accession numbers MN528387—MN528444. Other data analyzed during the current study are not publicly available due to the progress of analyzes of possible infections and co-infections with other pathogens but are available from the corresponding author on request.

### 4.7. Ethical Review

All participants were included after providing informed and written consent. They received their laboratory test results. HCV-infected participants were counseled and directed to seek. treatment in the public health facilities. This study was approved by the Committee for Ethics in Research of the Center for Tropical Medicine of the Federal University of Pará, in the state capital Belém, Brazil (CAAE: 37536314.4.0000.6287).

## 5. Conclusions

This study is unique in providing important epidemiological information on HCV infection in female sex workers in a difficult-to-access Brazilian region. The high prevalence of HCV infections, with a predominance of genotype 1 and presence of RASs, highlights the urgent need for better control and prevention measures regarding HCV diagnosis and treatment in an extremely vulnerable group.

## Figures and Tables

**Figure 1 pathogens-08-00236-f001:**
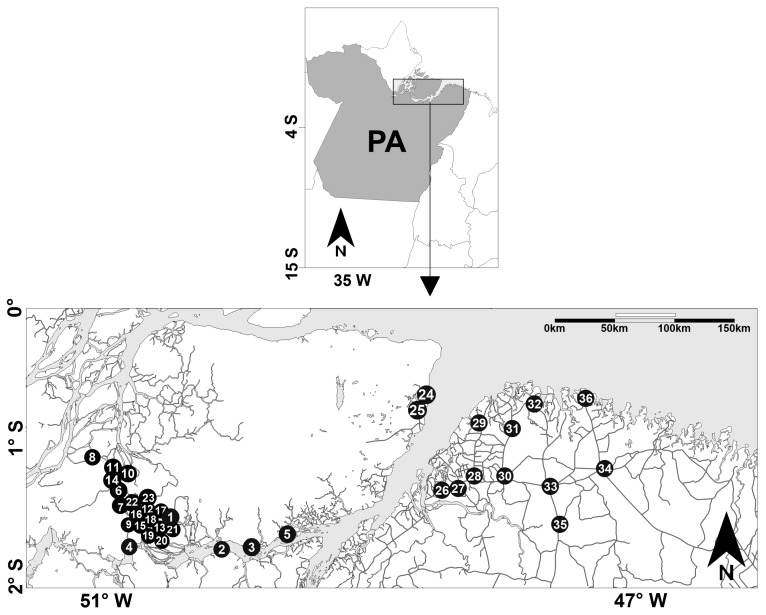
Sites at which biological samples and data were collected from female sex workers in the Brazilian state of Pará (PA), Amazon region (northern Brazil). Points 1 to 5 and 24 to 36 are cities (Breves (1), Bagre (2), Curralinho (3), Melgaço (4), São Sebastião da Boa Vista (5), Soure (24), Salvaterra (25), Ananindeua (26), Marituba (27), Santa Izabel do Pará (28), Vigia (29), Castanhal (30), Terra Alta (31), Marapanim (32), Santa Maria do Pará (33), Capanema (34), Sao Miguel do Guamá (35), and Salinópolis (36)) and from 6 to 23 are small riverside communities (Antônio Lemos (6), Capinal (7), São Francisco (8), Ramex (9), São Sebastião (10), Nossa Senhora de Fátima (11), Mainard (12), Intel (13), Campo Beija Flor (14), Zé Gama (15), Nova Canaã (16), Santa Cruz (17), Monte Tabu (18), São José (19), Corcovado (20)), Magebras (21), Bom Jesus (22), and Jupatituba (23)). More details can be found in Appendix A.

**Table 1 pathogens-08-00236-t001:** Demographic and socioeconomic characteristics of female sex workers associated with hepatitis C virus (HCV) exposure.

Characteristics	N Total	Anti-HCV+ (%)	Anti-HCV- (%)	*p* ***
Total	412	44 (10.7)	368 (89.3)	-
Age (years)				
18–24	172	22 (12.8)	150 (87.2)	0.49
25–30	167	15 (9.0)	152 (91.0)	
>30	73	7 (9.6)	66 (90.4)	
Color/Race (self-declaration)				
White	105	12 (11.4)	93 (88.6)	0.61
Brown (mixed race)	192	21 (10.9)	171 (89.1)	
Black	115	9 (7.8)	106 (92.2)	
Origin				
Born in the state of Pará	318	36 (11.3)	282 (88.7)	0.44
Not born in the state of Pará	94	8 (8.5)	86 (91.5)	
Sexual orientation				
Heterosexual	381	40 (10.5)	341 (89.5)	0.68
Same sex (including bisexual)	31	4 (12.9)	27 (87.1)	
Education Level				
Illiterate	72	11 (15.3)	61 (84.7)	0.17
Elementary school (incomplete/complete)	221	26 (11.8)	195 (88.2)	
High school (incomplete/complete)	110	6 (5.5)	104 (94.5)	
University (incomplete)	9	1 (11.1)	8 (88.9)	
Marital status *				
Married or co-habitating	33	5 (15.2)	28 (84.8)	0.39
Single, separated or widowed	379	39 (10.3)	340 (89.7)	
Monthly income (minimum wage) *				
≤1 **	307	38 (12.4)	269 (87.6)	0.15
2–3	79	5 (6.3)	74 (93.7)	
>3	26	1 (3.8)	25 (96.2)	

* Last 12 months; ** 1 minimum wage = R$ 890 (equivalent to US$ 220); *** *p*-value by Chi-square test.

**Table 2 pathogens-08-00236-t002:** Prevalence of infection and distribution of hepatitis C virus (HCV) genotypes among female sex workers in the Amazon region.

Marker (Laboratory Test)	Prevalence		95% CI
	Positive/Total	%	
HCV infection			
All (EIA+)	44/412	10.7	5.8–15.1
Active (EIA+ and PCR+)	32/412	7.8	2.5–12.5
Non-active (EIA+ Immuno Blot+ and PCR-)	12/412	2.9	0.0–8.2
Exposed (EIA+ and Immuno Blot+ or PCR+)	44/412	10.7	5.8–15.1
HCV genotypes			
Genotype 1	26/32	81.3	76.9–86.5
Genotype 3	6/32	18.7	14.7–23.4

**Table 3 pathogens-08-00236-t003:** Bivariate and multivariate analysis of risk factors for HCV exposure among female sex workers (FSWs) in the Amazon region.

Risk Factors	N Total	N Anti-HCV+	Bivariate *OR* (95% CI)	Multivariate *aOR* (95% CI)
Up to elementary school vs. high school or more	293	38	2.8 (1.2–6.8)	2.3 (1.3–6.3)
Up to one minimum wage vs. more than one minimum wage *	307	41	5.3 (1.6–17.4)	5.5 (1.7–16.8)
Illicit drug use (injectable or inhaled) vs. did not use illicit drugs *	130	34	9.7 (4.3–20.1)	9.4 (3.9–19.5)
Unprotected sex vs. protected sex **	156	42	35.8 (11.1–86.3)	32.1 (10.8–74.3)
More than five sexual partners vs. up to five sexual partners **	191	28	2.3 (1.2–4.5)	2.5 (1.3–4.2)
Condom exemption for clients paying extra vs. condom use for clients paying extra **	131	38	13.4 (5.3–31.4)	14.2 (4.9–28.4)
More than seven years working in the sex trade vs. up to seven years working in the sex trade	188	33	4.1 (2.1–8.3)	4.6 (1.8–7.6)
Changes in genitalia (wart, wound, and/or itching) vs. no changes in genitalia *	249	35	2.9 (1.3–5.9)	3.2 (1.2–6.0)
Did not perform medical/gynecological examination vs. performed medical/gynecological examination *	225	36	4.3 (1.9–9.5)	4.5 (1.7–8.2)

aOR: Adjusted odds ratio. * Last 12 months; ** Last 7 days.

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
