# Peer review of "Hepatitis C Virus among Female Sex Workers: A Cross-Sectional Study Conducted along Rivers and Highways in the Amazon Region"

_pathogens, 2019, doi:10.3390/pathogens8040236_

Round 1

Reviewer 1 Report

In this manuscript, the authors describe a cross-sectional study of the prevalence of hepatitis C virus (HCV) infection in female sex workers in the Amazon region of Brazil. Some of the strengths of the study are the large sample numbers (412) and the recruitment from regional areas rather than metropolitan areas. Further value adding is performed by genotyping and identification of potential drug-resistance polymorphisms. Prevalence of HCV infection was high and analysis of risk factors showed unprotected sex to be the major factor associated with transmission. This is important as current knowledge indicates that sexual transmission of HCV is a public health problem but most resources have been devoted to the study of men who have sex with men, particularly in those HIV-positive. Heterosexual transmission is believed to be low risk but a number of factors can enhance infection rates, many of which are described here.

There are a few items for the authors to attend to that can strengthen the manuscript.

In general, the manuscript is well written but there are a few typographical errors. Abstract (27): “found” for “fond”; Discussion (200): “haemophiliac” for “hemophilac”. Introduction. The Global Burden of Disease Study (2013) estimated that infection from HCV resulted in nearly 500,000 deaths in 2015. The authors should check their estimate. Throughout the manuscript, the authors describe drug-resistance mutations, however, these refer to changes associated with decreased response to protease inhibitors only. The new direct acting antivirals, which include drugs to other viral targets such as the viral polymerase, show a good response to HCV genotype 1. Additionally, I am unsure whether the naturally occurring amino acid variant at position N174 can be considered a true resistance-associated variant compared to the more common amino acid changes such as at Q80, R155, A156 and D168. I would prefer the authors refer to cDNA HCV as HCV RNA. If appropriate controls were used, then what was measured must be RNA. For Tables 1 and 3, the subheadings would be better expressed as anti-HCV+ and anti-HCV-.

Author Response

Dear Reviewer 1,

All suggestions have been accepted and are highlighted in blue on the manuscript. Other changes were made to increase the authenticity of the information (highlighted in green).

Thank you very much for your time and attention in contributing to a better manuscript!

The two typographical errors in the Abstract and Discussion have been fixed. New excerpts in the manuscript: “Previous studies found high prevalence of pathogens among female sex workers (FSWs) in the Amazon region…” and “In the Amazon region, the estimated prevalence of HCV infection ranges from 1% to 3% with higher rates being recorded in vulnerable groups, such as PWUDs (23.1%), patients undergoing hemodialysis (8.4%), and patients with hemophilia (48.4%)”. The authors modified the text to avoid incorrect estimates. New excerpt in manuscript: “The hepatitis C virus (HCV) infection is a major public health problem worldwide, an estimated 100 million people with serological evidence of HCV exposure, which may be associated with a growing trend in deaths per year”. The authors replaced the term “drug resistance mutations” for “resistance-associated substitutions (RASs)” and modified the text to not just refer to protease inhibitors. New excerpts in the manuscript: “Furthermore, resistance-associated substitutions (RASs) located in NS3, such as V36L, V55A, Q80L, and R155K for subtype 1a, and T54S, V55A, R117H, and D168G for subtype 1b, have already been identified in HCV-infected Brazilians [16]” and “Another urgent need is the provision of efficient treatment for HCV-infected people based on genotyping and RASs assessment”. The text was modified indicating only RASs with scientific reports. Furthermore, the possibility of N174S substitution to reduce teleprevir susceptibility, according to the program used, was reported, highlighting the absence of scientific report indicating the relationship of N174S with patients who did not respond to antiviral therapy. New excerpts in the manuscript: “The frequency of FSWs with RASs was 23.1% (6/26) for grazoprevir. Two RASs were detected: S122G (5/26, 19.2%) and Y56F (1/26, 3.9%). All RASs were detected in FSWs with subtype 1b. Interestingly, modification N174S was detected in six FSWs with subtype 1a. According to geno2pheno, N174S may indicate reduced susceptibility to telaprevir. In total, three modifications were found in the sequences of the NS3 viral protease coding region” and “In this study, this rate was even higher, 6 of 26 (23.1%) HCV-infected FSWs had RASs for grazoprevir. The S122G and Y56F substitutions were identified, and have already been described in the scientific literature [58-60]. On the other hand, there is no scientific report of the association of N174S with resistance in non-responders to viral therapy, but there are reports of its presence in patients infected with HCV subtype 1a [61]“. The replacement of "cDNA HCV" to "RNA HCV" and the changes in Tables 1 and 3 were also made. Check tables 1 and 3.

Reviewer 2 Report

Oliveira-Filho et al. determined the prevelance of HCV among female sex workers in the Amazon and also described the DAA resistance profile. With the role out of highly effective DAAs there is a large need to understand HCV prevelance in previously unidentified populations so as to be able to appropriately broaden the application of screening and treatment. This study indicated a population in Amazon with a relatively high prevalence and also that careful consideration of an appropriate DAA regime may be required to address naturally high DAA resistance profiles. The article was well written and of general interest to the HCV field. Below are some minor comments that should be addressed.

Pg1 line 27: change “fond” to “found”.

Pg 1 line 30: drug resistant mutations are usually referred to as resistance associated mutations (RASs).

Pg 2 line 52 – the new DAAs are pan-genotype.

Pg 3 line 98 – change “addressed” to “assessed”.

Author Response

Dear Reviewer 2,

All suggestions have been accepted and are highlighted in blue on the manuscript. Other changes were made to increase the authenticity of the information (highlighted in green).

Thank you very much for your time and attention in contributing to a better manuscript!

The correction of the term was made. The replacement of “fond” to “found” was made. New excerpt in manuscript: “Previous studies found high prevalence of pathogens among female sex workers (FSWs) in the Amazon region…” The correction of the term was made. The authors replaced the term “drug resistance mutations” for “resistance-associated substitutions (RASs)”. One of the new excerpts in the manuscript: “This study estimated the prevalence of HCV infection and associated risk factors, the frequency of HCV genotypes and resistance-associated substitutions (RASs) in this vulnerable group”. The authors provided more information about DAAs, including the suggestion given by referee. New excerpt in manuscript: “Thus, efforts were made to improve the efficacy and tolerability of HCV treatment, leading to the development of multiple direct-acting antivirals (DAAs) [9,10]. These drugs are molecules that target specific nonstructural proteins of the virus and results in disruption of viral replication and infection, regardless of HCV genotype”. The correction of the term was made. The replacement of “addressed” to “assessed” was made. New excerpt in manuscript: “In total, this study assessed 482 FSWs in different points of the sex trade in the Brazilian state of Pará in the Amazon region”.